# Inherently Area-Selective Atomic Layer Deposition of Manganese Oxide through Electronegativity-Induced Adsorption

**DOI:** 10.3390/molecules26103056

**Published:** 2021-05-20

**Authors:** Yi-Cheng Li, Kun Cao, Yu-Xiao Lan, Jing-Ming Zhang, Miao Gong, Yan-Wei Wen, Bin Shan, Rong Chen

**Affiliations:** 1State Key Laboratory of Digital Manufacturing Equipment and Technology, School of Mechanical Science and Engineering, Huazhong University of Science and Technology, 1037 Luoyu Road, Wuhan 430063, China; yichengli@hust.edu.cn (Y.-C.L.); miaogong@hust.edu.cn (M.G.); 2State Key Laboratory of Materials Processing and Die & Mould Technology, School of Materials Science and Engineering, Huazhong University of Science and Technology, 1037 Luoyu Road, Wuhan 430063, China; M202070873@hust.edu.cn (Y.-X.L.); simonzjm666@gmail.com (J.-M.Z.); ywwen@hust.edu.cn (Y.-W.W.); bshan@mail.hust.edu.cn (B.S.)

**Keywords:** area selective, atomic layer deposition, manganese oxide

## Abstract

Manganese oxide (MnO_x_) shows great potential in the areas of nano-electronics, magnetic devices and so on. Since the characteristics of precise thickness control at the atomic level and self-align lateral patterning, area-selective deposition (ASD) of the MnO_x_ films can be used in some key steps of nanomanufacturing. In this work, MnO_x_ films are deposited on Pt, Cu and SiO_2_ substrates using Mn(EtCp)_2_ and H_2_O over a temperature range of 80–215 °C. Inherently area-selective atomic layer deposition (ALD) of MnO_x_ is successfully achieved on metal/SiO_2_ patterns. The selectivity improves with increasing deposition temperature within the ALD window. Moreover, it is demonstrated that with the decrease of electronegativity differences between M (M = Si, Cu and Pt) and O, the chemisorption energy barrier decreases, which affects the initial nucleation rate. The inherent ASD aroused by the electronegativity differences shows a possible method for further development and prediction of ASD processes.

## 1. Introduction

Area-selective deposition (ASD), including area-selective atomic layer deposition (ALD), is considered a prospective way to downscale nano-electronics [1,2,3]. ALD takes advantage of self-limiting reactions to achieve conformal thin film growth with atomic-level thickness control [4]. The ALD reactions are extremely sensitive to surface chemistry due to the self-limiting nature [5]. Firstly, the activity of the precursors and steric hindrance from the ligands affect the density of chemisorption [6]. Meanwhile, the type and density of surface-active sites have an important role in the growth rate of target materials [7]. It is generally considered that the O*, H* and –OH are the active sites for the precursors’ chemisorption [8,9,10]. In some conditions, the byproducts or impurities may poison the surface and prevent additional adsorption of the precursors [11,12]. Moreover, a surface with catalytic activity can promote ALD growth [13,14]. When the driving force of nucleation is between the Gibbs free energy barriers of two different surfaces, the nuclei form rapidly on a surface with a low-energy barrier rather than on the other. Thus, ideal area-selective ALD can be realized through nuclei inhibition on the nongrowth areas. The selective deposition is quite complicated and relies on the coupled parameters, including substrates, precursors, temperature, pressure and aspect ratio dependency. Prior reports showed that the selectivity could be obtained by chemical modification, including surface passivation with polymers, self-assembled monolayers (SAMs), inhibitors and so on [15,16,17]. The above methods achieve high selectivity for nano-feature patterning but have limitations in long passivation time and complex steps. Thus, inherently selective deposition has been investigated to relax process complexity [18]. Up to now, four types of ASD have been reported, including DoD, DoM, MoD and MoM (M = metal, D = dielectric). Among them, selective deposition of oxides on metal can be used as barrier or capping layers to improve device performance.

ALD-MnO_x_ has been studied and utilized to reduce the overall interconnect resistance in microelectronics [19]. It was also applicable to energy storage and conversion [20], magnetic devices [21] and so on. Previous studies demonstrated that the manganese oxide was fabricated by ALD using bis(ethylcyclopentadienyl) manganese (Mn(EtCp)_2_) and water (H_2_O) [22,23] or tris(2,2,6,6-tetramethyl-3,5-heptanedionato) manganese (Mn(thd)_3_) and ozone (O_3_) [24] as precursors. The MnO_x_ layer was conformally deposited in the high aspect-ratio structures, and the ~1 nm thin film showed great diffusion barrier properties [25]. Usually, the deposition took place on the entire surface and needed auxiliary etching to obtain nanopatterns. Kawasaki et al. found that the selective deposition of MnO_x_ existed between Cu and low-k SiCOH films, and the nucleation delay on SiCOH films is just a few cycles [26]. Moreover, they found that the native oxide layer on Cu is in favor of an ALD reaction compared with non-oxidized Cu. Phuong et al. discovered that plasma-treated and as-received SiOC substrates had different deposition rates of MnO_x_ [27]. It was found that moisture was adsorbed on the matrix after plasma treatment and acted as a catalyst for MnO_x_ deposition. The chemical bond between the deposited oxide and the substrate is M–O–Mn (M = metal, metalloid), and the chemical state of the surface oxygen group is the key issue. Previous reports established a relationship between the type of oxygen species and the nucleation rate [28]. Therefore, it is of great significance to study the surface chemistry and reaction mechanism to obtain high selectivity.

In this article, MnO_x_ is deposited on metal/SiO_2_ patterns using Mn(EtCp)_2_ precursors with H_2_O as co-reactant. The X-ray photoelectron spectroscopy (XPS) shows that the Mn (II) precursors are oxidized, and the substrates are reduced after the ALD process. Additionally, the surface oxygen group plays an important role in chemisorption. The selectivity is quantified by EDS mapping, and the selectivity is optimized by adjusting deposition temperature. Meanwhile, it was found that selectivity has a relationship with the electronegativity differences. The differences in electronegativity decrease in the order of Si–O (1.54), Cu–O (1.44) and Pt–O (1.16). On the contrary, the deposition rate is negatively related to the electronegativity differences. It is conducive to the formation of M–O–Mn (M = metal, metalloid) bonds in areas with a low difference in electronegativity.

## 2. Results

### 2.1. Characterization of Selectivity

The schematic diagram of MnO_x_ deposited on metal/SiO_2_ patterns is shown in Figure 1a. The patterns consist of 60 μm metal-capped lines. There are sharp boundaries between the metal and exposed SiO_2_/Si areas (Appendix A from Appendix A). After 50 ALD MnO_x_ cycles, the amplified border areas are characterized with SEM and EDS. Figure 1 shows EDS mapping of Pt and Mn on Pt/SiO_2_ patterns at 80, 125 and 215 °C, respectively. The green and white points represent Pt and Mn elements, respectively. It is found that the density of white points (Mn) on the Pt surface is higher than that on the SiO_2_/Si surface, indicating that MnO_x_ grows faster on the Pt surface. The white points decrease on the nongrowth SiO_2_ regions with increasing deposition temperature, indicating that the selectivity is improved. The temperature adjustment can enlarge the differences in the deposition rates on different substrates. The quantitative values of selectivity and the corresponding selectivity window are presented in the following parts.

### 2.2. Surface Composition of MnO_x_ Deposited on Different Substrates

The substrates’ surfaces affect nucleation behaviors of selective deposition and subsequent quality of target materials. To identify the influences of surface species during ALD, the composition of MnO_x_ deposited on Pt, Cu and SiO_2_ substrates are analyzed by XPS. Firstly, the appearance of Mn peak indicates that MnO_x_ is grown on three kinds of substrates. The Mn 2p spectrum (Figure 2a) shows two peaks corresponding to the Mn 2p_3/2_ and Mn 2p_1/2_ core levels centered at binding energies of ~653.0 eV and ~642.0 eV, respectively [29,30]. After the spectra fitting analysis, the Mn 2p_3/2_ spectra are separated with four peaks located at around 641.1–641.3 eV, 642.1–642.6 eV, 644.0–644.3 eV and 646.5–647.0 eV, which correspond to Mn^2+^, Mn^3+^, Mn^4+^ and Mn^2+^ satellite bonds, respectively. It is found that there are three valence states of Mn element on Pt and SiO_2_ surface, while there is no tetravalent Mn on Cu surface.

The Mn(EtCp)_2_ precursor is a reducing agent due to its unsaturated –EtCp ligands. Thus, the MnO_x_ film presents higher valences due to oxidation during the ALD process. The percentage of tetravalent Mn on Pt is higher than that on the SiO_2_ surface. This is perhaps because of the catalytic activity of Pt. Previous studies attributed ASD of metal oxides (such as FeO_x_, CeO_x_) on Pt/SiO_2_ patterns to catalytic activation of oxygen that increase nucleation rate on Pt [31]. Due to the high activity of surface oxygen species on Pt, it is easier for the precursor’s adsorption and decomposition, thus leading to high selectivity.

To analyze the role of surface oxygen species, the SiO_2_/Si, Pt and Cu substrates before and after MnO_x_ deposition are measured with XPS and presented in Figure 3. All of the substrates are reduced in a 10% hydrogen mixture at 125 °C, and then the substrates are transferred into the ALD reactor to deposit MnO_x_. It is found that Pt and Cu substrates are partially oxidized before the ALD process, although the substrates are already reduced. Hence, surface oxygen species will affect the nucleation behaviors during selective deposition. After 50 ALD cycles, Pt is slightly reduced, while the Cu substrate is obviously reduced (Figure 3b,c). The electron transfer occurs between the substrate and the Mn precursors, which results in the reduction of the substrate [30]. The surface oxygen species act as active sites for MnO_x_ deposition. The rapid disappearance of CuO may lead to a great decrease of active sites and the weakening of the electron-stripping ability of the substrate, resulting in the absence of tetravalent manganese. Additionally, the O 1s XPS spectrum of SiO_2_/Si, Pt and Cu substrates are presented. After the MnO_x_ deposition, the content of lattice oxygen is more than that of the bare substrate without deposition. The increase of lattice oxygen may contribute to the Mn–O bonds after deposition.

### 2.3. Selectivity Window of MnO_x_

To obtain high selectivity, it is necessary to suppress nuclei on the nongrowth areas. To quantify the selectivity, Equation (1) is as follows [32]:(1)$$\mathrm{Selectivity}=\frac{\theta_{GA}-\theta_{NGA}}{\theta_{GA}+\theta_{NGA}}$$
where *θ_GA_* and *θ_NGA_* are the amount of target materials deposited on the growth and nongrowth areas, respectively. In the experiments, *θ* is simplified as the number of white points (Mn) per unit area in EDS mappings. The larger the number of white points of Mn, the more MnO_x_ deposited on the substrate. Thus, EDS mapping can be used as a semi-quantitative method to acquire the values of selectivity.

Through mathematical counting of Mn EDS mapping in Figure 1, the quantitative selectivity values are obtained. The selectivity on Pt/SiO_2_ patterns increases from 0.39 to 0.5 through increasing temperature from 80 to 125 °C. Then, the selectivity slightly decreases to 0.48 at 215 °C (Figure 4a). The selectivity between Cu and SiO_2_ surface is also analyzed, and there is almost no selective growth.

To study the relationship between the selectivity and the ALD temperature window, the growth rate is evaluated at the same temperature range (Figure 4b). As the deposition temperature increases from 80 to 125 °C, the growth rate of Mn oxides decreases quickly (50 ALD cycles are exploited). Further increasing the deposition temperature has minimal influence on film thickness, indicating that the temperatures ranging from 125 to 215 °C are within the ALD window. It is found that high selectivity can be obtained only in the ALD window. The Mn precursors may condense at 80 °C, which causes weak selectivity. 

The MnO_x_ film thickness increases linearly on SiO_2_ substrate after 50 cycles (Figure 4c). The steady growth rate is 0.01 nm/cycle at 125 °C. The slow nucleation on the SiO_2_ regions is thought to be beneficial to achieve selective deposition. The growth rate within 50 ALD cycles is also presented in Figure 4d. The first 50 ALD cycles show two growth stages: ~10 cycles of nucleation incubation and a rapid growth stage indicating the acceleratory growth of nuclei islands. After 50 cycles, the surface of the SiO_2_ substrate is almost covered, and the nuclei islands begin to coalesce. The growth rate remains stable when further increasing ALD cycles. The surface gradually changes from SiO_2_ to MnO_x_ layer after approximately 50 cycles, and the deposition rate decreases. When all the surfaces are completely covered with MnO_x_, the substrate differences between Pt, Cu and SiO_2_ are limited. For MnO_x_ grown on ultrathin Pt films, it is found that the film thickness is ~2.8 nm at 50 ALD cycles, which is much higher than that on the SiO_2_ substrate. The nucleation stage is also studied by examining the surface morphology of ALD MnO_x_ ultrathin films with SEM and AFM (Appendix A from Appendix A). SEM image shows that there are a lot of nuclei islands on SiO_2_ at 50 ALD cycles. After 400 ALD cycles, the particle size increases from ~9.1 to ~16.7 nm. AFM images show that initial MnO_x_ deposition increases the surface roughness, then it decreases slowly from ~0.80 to ~0.64 nm with increasing ALD cycles (see Appendix A from Appendix A).

The GPC is also checked by changing the pulse time of the precursors. The critical time of saturated adsorption of Mn precursors and water are ~2 and ~3 s, respectively (Appendix Aa,b from Appendix A). After the minimum dose time, the thickness maintains at 0.9~1.0 nm with increasing dosing time. It is found that the purge times longer than 30 and 60 s are enough to remove excess Mn(EtCp)_2_ and H_2_O, respectively. The long pulse time of Mn(EtCp)_2_ (3s) and H_2_O (3s) in this work are to ensure sufficient adsorption and decomposition. The above results exhibit good self-limiting character, which lays a foundation for selective growth.

## 3. The Origin of Selective Growth 

It is considered that ASD is mainly originated from competition between surface chemisorption, decomposition and so on [33]. To elucidate the origin of selective growth, the density functional theory (DFT) and nudged elastic band (NEB) are adopted to calculate the decomposition energy and energy barriers of ALD precursors on Pt, Cu and SiO_2_ surfaces, as shown in Figure 5a. On the SiO–OH surface, the following reactions between Mn(EtCp)_2_ and surface are considered as Equation (2) [34,35,36]:Mn(EtCp)_2_ + ‖–OH=‖–O–Mn(EtCp) + H–EtCp(2)

According to the previous study, the following decomposition reactions on the surface of Pt and Cu are considered as Equation (3) [37,38]:Mn(EtCp)_2_ + ‖–O=‖–O–Mn(EtCp) + ‖–O–EtCp(3)

Firstly, the adsorption energy of Mn precursor on three different substrates is analyzed. On the surface of SiO_2_, Mn(EtCp)_2_ is adsorbed on the –OH sites, and the adsorption energy is −1.01 eV. In the decomposition step, the EtCp group combines with the H atom of –OH to form H-EtCp, and the decomposition energy barrier is 2.64 eV. Mn(EtCp)_2_ is adsorbed on Pt and Cu surfaces with EtCp ligand facing the surface, and the adsorption energies are −1.66 and −2.37 eV, respectively. The Mn-EtCp part slips from the EtCp group to the Pt and Cu surfaces, and the decomposition barriers are 0.92 and 2.55 eV, respectively. 

The above results reveal that the growth of Mn(EtCp)_2_ has priority in the order of Pt > Cu > SiO_2_. Then, the partial density of states (PDOS) of precursors adsorbed on Pt and Cu surface is analyzed (Figure 5b). From the PDOS diagram, the overlap of C_p orbital and Pt_d orbit indicates that a strong chemical bond is formed between the Pt surface and the precursors. In contrast, such an orbital overlap is absent on the Cu surface. The preferential growth of MnO_x_ on the Pt surface may come from the favorable adsorption and decomposition process of Mn precursors.

To further explore the effect of substrates, the relationship between the selectivity and differences in electronegativity between M and O is presented (Figure 6a). The Pauling electronegativity of Pt, Cu, Si and O are 2.28, 2.00, 1.90 and 3.44, respectively [39]. The differences in electronegativity between M (M = Pt, Cu and Si) and O decrease in the order of Si–O (1.54), Cu–O (1.44) and Pt–O (1.16). It is found that selectivity is negatively correlated with the differences in electronegativity. Smaller differences in electronegativity between M–O may lead to weaker bonding energy [40,41], which may influence the chemisorption of Mn precursors on Si–O, Cu–O and Pt–O surfaces. The absolute value of adsorption energy rises with the order of Si–O, Cu–O and Pt–O from the DFT calculations, and the reaction barrier decreases with the same order. The differences in electronegativity may also become a supplemental explanation of selectivity for some precursors on certain substrates. The origin of selective deposition is the differences in chemisorption between two surfaces. At the nucleation stage, the precursors are chemically adsorbed on the surface oxygen groups to form M–O–Mn bonds. The bonding energy of M–O, which is affected by the electronegativity, thus affects chemical adsorption. Smaller differences in electronegativity between M–O may lead to weak bonding, which is beneficial to reduce the reaction barrier.

## 4. Materials and Methods

### 4.1. Substrate Preparation

Cu and Pt films were deposited by sputtering onto Si(100) wafers using an argon-based plasma. The film thickness is a few nanometers. The surface roughness of the Pt and Cu substrates are less than 5 nm. Si (100) substrates, with a layer of SiO_2_ around 20 Å thick, were cleaned with acetone and deionized water. The Pt and Cu samples were annealed in hydrogen to remove the oxide layers as much as possible. If they were not immediately used after cleaning, the substrates are stored in the argon atmosphere until use.

### 4.2. Growth Conditions

The ALD reactions were performed in a custom hot-walled ALD reactor (Material Design and Nano-manufacturing center @ HUST, Wuhan, China) at 50−250 °C, and 50 Pa of argon pressure provided by an 80 Sccm continuous argon purge (99.999%, Prepurified). ALD of MnO_x_ was carried out using (a) Mn(EtCp)_2_ (>98%, Aimou Yuan, Nanjing, China) and (b) H_2_O (ultrapure deionized water). The temperature of the precursor’s container was controlled at 100 °C, in which argon was diverted over the headspace of the liquid precursors during dosing. The temperature of the pipeline was 20 °C higher than that of the container to prevent condensation of Mn precursors. The inset of Appendix Aa (from Appendix A) shows the typical half-reactions process of Mn(EtCp)_2_ and H_2_O dosing. The base pressure of the ALD chamber was about ~113 Pa, the peak pressure of the Mn(EtCp)_2_ and H_2_O dose were tuned to a level of ~20 and ~7 Pa above the base pressure, respectively. The ALD reactions were carried out in a circulate of Mn(EtCp)_2_, purge, H_2_O and purge sequence, with a typical time of 3, 30, 3, and 60 s.

### 4.3. Characterization of MnO_x_

The film thickness on SiO_2_ was measured by a spectroscopic ellipsometer (SE, M-200X, J. A. Woollam Co., Inc., Lincoln, NE, USA). Additionally, MnO_x_ thickness on ALD Pt was acquired by SE after 50 ALD cycles at 125 °C. The wavelength of light ranged from 250 to 1000 nm. The software was Complete EASE, and a modified Cauchy model was used to fit the ellipsometer data. Patterned Pt/SiO_2_ and Cu/SiO_2_ substrates were observed by optical microscope. The surface morphology was analyzed by SEM (attached EDS, SU3900, HITACHI, Tokyo, Japan) and AFM (Agilent 5500, Santa Clara, CA, USA) for 0, 50 and 400 MnO_x_ ALD cycles. EDS mappings of platinum and manganese were measured on Pt/SiO_2_ patterns after 50 ALD MnO_x_ cycles at 80, 125, and 215 °C. The surface composition was evaluated by X-ray photoelectron spectrograph (XPS, AXIS-ULTRA DLD-600 W, Shimadzu-Kratos Co., Manchester, UK) with an Al Kα X-ray (hν = 1486.6 eV) after 50 ALD cycles on Cu, Pt and SiO_2_ substrates. All the spectrum were calibrated using C 1s (285 eV). The peaks were separated by XPS peak software, and the background was removed by Shirley type. The peak areas of Mn^2+^, Mn^2+^ satellite, Mn^3+^ and Mn^4+^ within Mn 2p_3/2_ were two times higher than that of Mn 2p_1/2_, respectively. The ratio of Lorentzian–Gaussian was 20%. The peak area represented the content of different bonds.

### 4.4. DFT Calculation

DFT calculations were carried out using the first-principles plane-wave pseudopotential formulation implemented in the Vienna ab-initio Simulation Package (version 5.4.4, University of Vienna, Vienna, Austria). The exchange−correlation functional was in the form of Perdew−Burke−Ernzerhof with the generalized gradient approximation [42]. Van der Waals interactions were also accounted to be based on experience. The cutoff energy was 400 eV on the basis of the plane-wave. A k-mesh of 3 × 3 × 1 was applied to ensure the energy convergence to 1 meV, and the residual force acting on each atom is less than 0.05 eV/Å. A 5 × 5 × 4 Pt(111) slabs model, 5 × 5 × 4 Cu(111) slabs model and 13 layers hydroxylated SiO_2_ slabs model was built to resemble the Pt, Cu and SiO_2_ surface on the experiment, respectively. The adsorption energy (E_a_) of the precursor on the Pt surface was defined by the Equation (4) [38]:E_a_ = E*_ads_ − E* − E_Mn(EtCp)2_(4)
where E*_ads_ is the total energy of the slabs; E* and E_Mn(EtCp)2_ are the energies of the slabs and precursor molecule, respectively. The decomposition energy (E_d_) is calculated by the Equation (5):E_d_ = E*_dec_ − E* − E_Mn(EtCp)2_(5)
where E*_dec_ is the total energy of the surface with the precursor decomposed. The nudged elastic band (NEB) was used to search the minimum energy path (MEP) of the precursor dissociation on the Pt surface and locate the transition state between two local minima. Six intermediate images were interpolated between the adsorbed state and decomposed state.

## 5. Conclusions

In this work, MnO_x_ films are fabricated with ALD using bis(ethylcyclopentadienyl) manganese (Mn(EtCp)_2_) and H_2_O. Inherently selective growth is demonstrated on patterned Pt/SiO_2_ and Cu/SiO_2_ substrates. It is found that MnO_x_ preferential deposits on Pt, then followed by Cu and SiO_2_. The selectivity improves with increasing deposition temperature within the ALD window. Meanwhile, it is found that the calculated selectivity has a negative relationship with enlarging the electronegativity differences between M (M = Si, Cu and Pt) and O. The origin of selective deposition is the differences of chemisorption between two surfaces. The bonding energy of M−O is affected by the electronegativity, thus affects chemical adsorption. Smaller differences in electronegativity between M−O may lead to weak bonding, which is beneficial to reduce the reaction barrier. The differences in electronegativity may also become a supplemental explanation of selectivity for some precursors on certain substrates, which shows a possible method the further development and prediction of selective ASD processes.

## Figures and Tables

**Figure 1 molecules-26-03056-f001:**
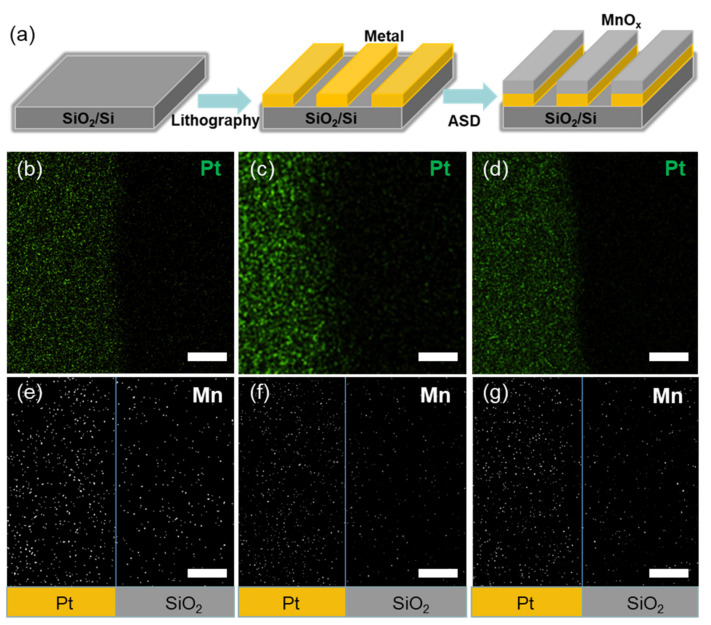
(**a**) A sketch of the area-selective ALD on metal/ SiO_2_ substrates patterned by lithography. EDS mappings of platinum and manganese on Pt/SiO_2_ patterns after 50 ALD MnO_x_ cycles, respectively. (**b**) Platinum and (**e**) manganese at 80 °C, (**c**) platinum and (**f**) manganese at 125 °C, and (**d**) platinum and (**g**) manganese at 215 °C. The scale bar is 200 nm.

**Figure 2 molecules-26-03056-f002:**
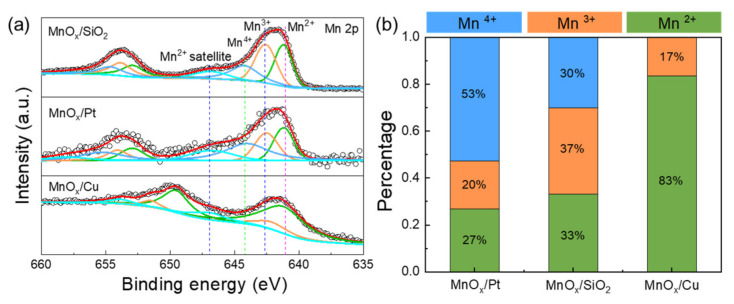
(**a**) Mn 2p XPS spectra of 50 ALD cycles on Cu, Pt and SiO_2_ substrates. (**b**) A comparison of bond information regarding the elements of Mn for the ALD MnO_x_ obtained from the XPS measurements.

**Figure 3 molecules-26-03056-f003:**
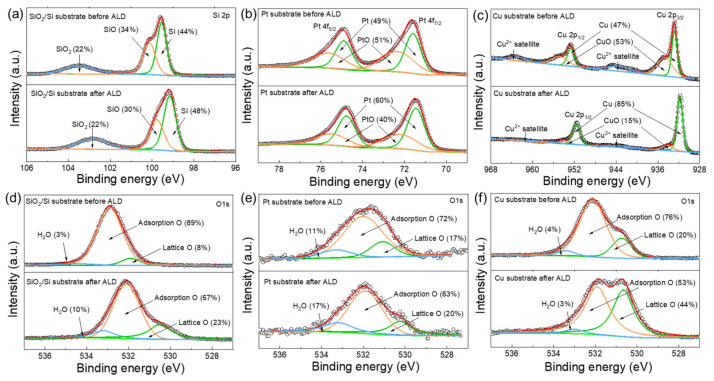
(**a**) Si 2p spectrum of SiO_2_/Si substrate before and after MnO_x_ deposition. (**b**) Pt 4f spectrum of Pt substrate before and after MnO_x_ deposition. (**c**) Cu 2p spectrum of Cu substrate before and after MnO_x_ deposition. O 1s XPS spectrum of (**d**) SiO_2_/Si substrate, (**e**) Pt and (**f**) Cu substrates, respectively.

**Figure 4 molecules-26-03056-f004:**
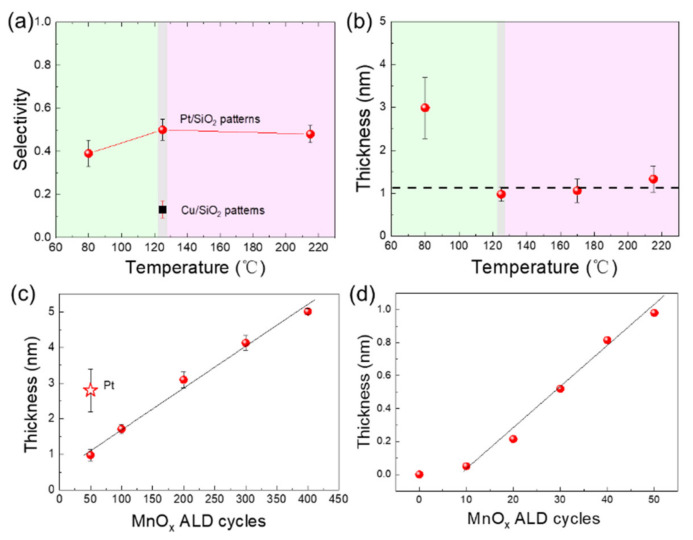
(**a**) The selectivity values as a function of the deposition temperature after 50 ALD MnO_x_ cycles. The film thickness with respect to (**b**) deposition temperature and ALD cycle number (**c**) after 50 and (**d**) within 50 cycles on SiO_2_ substrate. The inset point in (**c**) shows the film thickness on the Pt surface at 125 °C after 50 ALD cycles.

**Figure 5 molecules-26-03056-f005:**
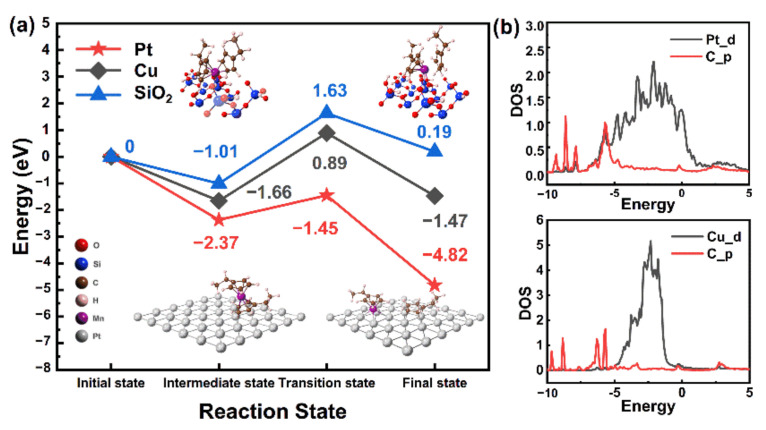
(**a**) Calculated energy diagrams of Mn precursors on the Pt, Cu and SiO_2_ substrates and (**b**) a PDOS diagram of the adsorption of Mn precursors on Pt and Cu surfaces.

**Figure 6 molecules-26-03056-f006:**
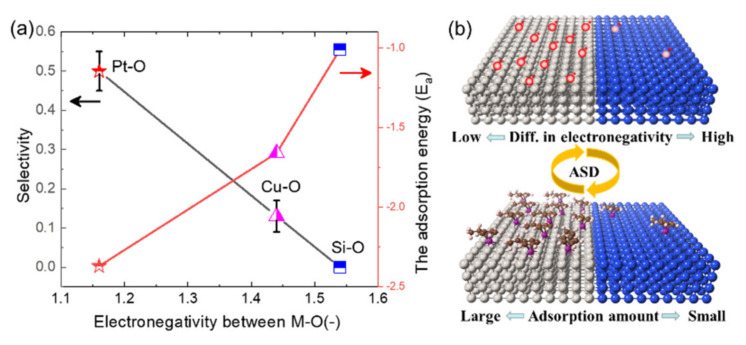
(**a**) The selectivity values (black line) and the adsorption energy (red line) plotted versus the differences in electronegativity of M–oxide (M = Si, Cu, Pt) underlayers. (**b**) Schematic for the adsorption amount of Mn(EtCp)_2_ on the patterned substrates with different D-value in electronegativity.

## Data Availability

Not applicable.

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
