# Peer review of "Inherently Area-Selective Atomic Layer Deposition of Manganese Oxide through Electronegativity-Induced Adsorption"

_molecules, 2021, doi:10.3390/molecules26103056_

Round 1

Reviewer 1 Report

This manuscript interestingly demonstrates that AS-ALD of MnOx can be achieved on various metal/dielectric patterns by differences in electronegativity in order of Si-O, Cu-O, Pt-O. The results they present are supported very well. Therefore, the reviewer suggests that this work deserves publication with addressing a few questions as below.

1) on page 3, line 112 : The authors stated that “There’s no 4-valent oxidation state of manganese on Cu surface. It’s obvious that the amount of Cu-O bonds is higher than that on Pt and SiO2 surface. The reason may come from competitive oxidation of Cu and Mn precursors during ALD reaction. Please provide any reference associated with this statement.  

2) on page 3, line 119 : Have the authors tried ALD MnOx deposition after reducing surface oxidation? Since the authors stated the role of oxygen species for selective deposition, it seems interesting to check. Also, H-terminated Si can be the option to check.

3) on pate 4, line 153 : The growth rate within 50 cycles seems to be a bit higher (0.025 nm/cy?) than that after 50 cycles (0.1 nm/cy). What makes such difference?

Reviewer 2 Report

Review manuscript rong chen

The manuscript provides interesting data on the nucleation behaviors of MnOx on different substrates. The authors prepared MnOx layers on metal/SiO2 patterns, and the selectivity has been quantified by EDS and XPS techniques. The authors claim that the selectivity of the deposition is (negatively) correlated with the difference in electronegativity of the substrates, and that the growth of Mn(EtCp)2 has the priority in the order of Pt > Cu > SiO2. This is interesting, but this finding should be put in comparison/correlation with other key parameters of surface selectivity already demonstrated, such as their catalytic activity. In fact, the catalytic activity shows the same Pt > Cu > SiO2 order, and this may prevail over the electronegativity. Overall, the paper is attractive and properly presented, and, once this issue tackled and the manuscript revised, I believe it would be a good fit for this special issue Self-Assembled Materials and Bottom-Up Fabrication of the Molecules journal.

Although the manuscript is interesting – some changes need to be carried out and properly addressed prior to publication.

  • The English grammar is not optimal, and the reader has the feeling that it has been written too quickly. For example, already the first sentence of the abstract is not well written. The authors should precise the nature of MnOx (thin films, nanolayers, nanomaterials…), and the sentence should not end with “etc.”; or in line 106, « This perhaps because », the « is » is missing ; or in line 277, « a novel self-limiting growth of MnOx are presented », « are » should be replaced with « is ».

The manuscript should be reread and revised in order to correct the grammar, improve the English and to make the reading more pleasant.

  • In the Introduction, the authors correctly indicate that “The ALD reactions are extremely sensitive to the surface chemistry. » It would be good to add a few sentences here explaining why (chemisorption of precursors, self limiting chemistry, catalytic activity etc).

  • In Figure 1, the differences of “the density of white points » that are supposed to back up the selectivity differences is not clearly appearing in the EDS mappings, even when zooming in. This is a key issue, that should be resolved. The mathematical analysis of the EDS, and/or the ellipsometry data could be added there for example.

  • The author used XPS to characterize the deposited layers. The data presented is clear and the peaks seem to have been properly deconvoluted. One question raising is the way how the background has been substracted. In fact, in Figure 2a, the background seems to have been removed differently, and no details are given, nor in the text nor in the experimental section. The authors should precise the XPS parameters they used and how they fitted the extracted data, including the removal of the background (Shirley, Tougaard?).

  • Again on the XPS data, one detail to modify is the y axis of Figures 2c and 2d, as a percentage should go from 0 to 100. Currently the ratio is shown (0 to 1). In addition, it would be nice to add the XPS data of the bare substrates.

  • The main issue of the paper is the following: the authors claim that the selectivity of the deposition is (negatively) correlated with the difference in electronegativity of the substrate, and that the growth of Mn(EtCp)2 has the priority in the order of Pt > Cu > SiO2. This is interesting, but this finding should imperatively be put in comparison/correlation with other key parameters of surface selectivity already demonstrated, such as their catalytic activity. See and compare for example the papers where were core/shell catalytic nanoparticles have been prepared using selective ALD (Weber et al, Nanotechnology 26 (9), 094002 (2015)). In fact, it has been shown that the catalytic activity of noble metals allows for the selectivity (for certain precursor chemistries) to take place, and a comparison with this electronegativity findings should be made. In fact, the catalytic activity shows the same Pt > Cu > SiO2 order, and the catalytic activity may be much more important than the electronegativity for ALD to take place. This matter should be tackled and discussed consequently. One good experiment could be done, with substrates showing different electronegativities but no catalytic activity.

  • The characterization of MnOx films in the Experimental part should be more detailed.

Round 2

Reviewer 2 Report

The authors properly answered the issued I raised, and i believe the paper can be published now.